# CH-CEMS: A Chinese Multi-Concept Benchmark Dataset Towards Explainable Multi-Modal Sentiment Analysis

## Abstract

Explainable Multimodal Sentiment Analysis (EMSA) is a booming research area aimed at advancing robust and faithful multimodal language understanding. Recent explainable datasets and methods based on multimodal large language models (MLLMs) have introduced a new paradigm that produces chain-of-thought–style explanations within affective computing. However, high-quality data resources for EMSA remain scarce, largely because annotating reliable reasoning cues is costly and difficult. To address this gap, we introduce CH-CEMS, the first multimodal sentiment dataset for explainable multimodal sentiment analysis. It contains 3,715 curated video segments with polarity and intensity annotations. In addition, we annotate three semantic concepts for each sample (i.e., speaking style, tone of voice, and facial expression), which serve as explicit reasoning cues to enable process-level supervision. To fully leverage these concept cues, we propose a concept-guided reinforcement learning framework with Group Relative Policy Optimization (GRPO) for MLLMs, in which concept-level supervision explicitly constrains cross-modal semantic relations and guides the model to infer sentiment from verifiable concepts. We further establish baselines with state-of-the-art multimodal machine learning methods and MLLMs via zero-shot inference and supervised fine-tuning. Experiments show that MLLMs outperform feature-based methods, typically by 4–12% in accuracy for three-class sentiment analysis, and that our concept-guided GRPO yields a further 8.5% improvement, even surpassing closed-source model such as GPT-5. We believe CH-CEMS and the benchmark will facilitate future research on explainable multimodal sentiment analysis. The dataset and codes are avaliable for use at `https://anonymous.4open.science/r/CH-CEMS-C34F`.

## 1 INTRODUCTION

Multimodal Sentiment Analysis (MSA), which integrates text, speech, and vision to overcome the limitations of uni-modal text by leveraging complementary multimodal information for more robust sentiment understanding, holds significant importance within the multimodal language understanding. The progress of MSA has been largely driven by benchmark datasets such as CMU-MOSI (Zadeh et al., 2016), and CMU-MOSEI (Zadeh et al., 2018), which provide essential resources for the training and systematic evaluation of multimodal models. Furthermore, in order to achieve friendly representations for multimodal fusion , datasets with modality-specific sentiment annotations are proposed such as CH-SIMS (Yu et al., 2020) and CH-SIMS2.0 (Yu et al., 2023). Building upon these datasets, prior studies have mainly emphasized intra-modal representation learning and inter-modal fusion. Though these approaches have achieved strong performance, they remain confined to feature-level modeling and have limited capacity to capturing high-level and diverse multimodal semantics, making it difficult to reason over explainable and verifiable cues under traditional modeling frameworks and task formulations. In real-world scenarios, sentiment polarities across modalities often diverge, and multimodal sentiment cannot be reduced to a simple aggregation of unimodal predictions. Existing methods typically address such modality inconsistency through feature weighting or selection (Li et al., 2024; 2025; Zhao et al., 2025b), but these strategies are still heuristic adjustments that lack semantically grounded modeling of inter-modal relations and consistency. To move beyond such limitations, a promising direction is to construct multimodal datasets enriched with explainable and verifiable semantic cues, and to leverage these cues as context-dependent references for reasoning.

Recently, multimodal large language models (MLLMs) have been applied to MSA (Jim et al., 2024; Yang et al., 2024b; Zhang et al., 2025a). A prevailing paradigm formulates the task in an autoregressive manner with MLLMs, where multimodal inputs are implicitly fused through alignment or attention in latent feature spaces, and optimized via supervised fine-tuning (Luo et al., 2025; Zhang et al., 2025b). Nevertheless, such approaches rarely impose se-

mantic constraints or consistency mechanisms, leaving the reasoning process untraceable. Large-scale MLLMs have demonstrated the capability to extract high-level semantic cues across modalities and underpin reliable multimodal reasoning (Wang et al., 2025b). While recent multi-stage or agent-style pipelines somewhat instruct models to understand such concepts to help reason (Huang et al., 2024; Fei et al., 2023; Zhang et al., 2025c), explicit training of the reasoning capability is underexplored. Meanwhile, following the advent of DeepSeek R1 (DeepSeek-AI, 2025), reinforcement learning (RL)-based training, exemplified by Group Relative Policy Optimization (GRPO) (Shao et al., 2024), has been shown to improve explanatory behavior in large language models and substantially enhance semantic-level reasoning in multimodal settings (Wang et al., 2025b). In parallel, researchers have also made effective attempts toward more explainable affective computing. The EMER dataset provides explainable reasoning data for emotion recognition (Liu et al., 2023a), and the MERR dataset extends this line by offering multimodal emotion description and benchmarks (Wu et al., 2023). Building on these resources, Emotion-LLaMA (Zhang et al., 2024) adopts multimodal instruction tuning to supervise explanation generation, demonstrating that explicit reasoning supervision can significantly improve both recognition accuracy and explainability. Furthermore, R1-Omni (Zhao et al., 2025a) optimizes an omni-modal model via reinforcement learning with verifiable rewards (RLVR) and GRPO, thereby substantially improving the model's reasoning capability and explainability. However, most explainable MSA approaches instantiate explanations as chain-of-thought (CoT) rationales, whose faithfulness is debated and which are generally unsupervised at the process level, leaving hallucinated reasoning uncorrected. Moreover, to the best of our knowledge, explainable multimodal datasets for sentiment analysis with process-level supervision and verifiable reasoning remain scarce.

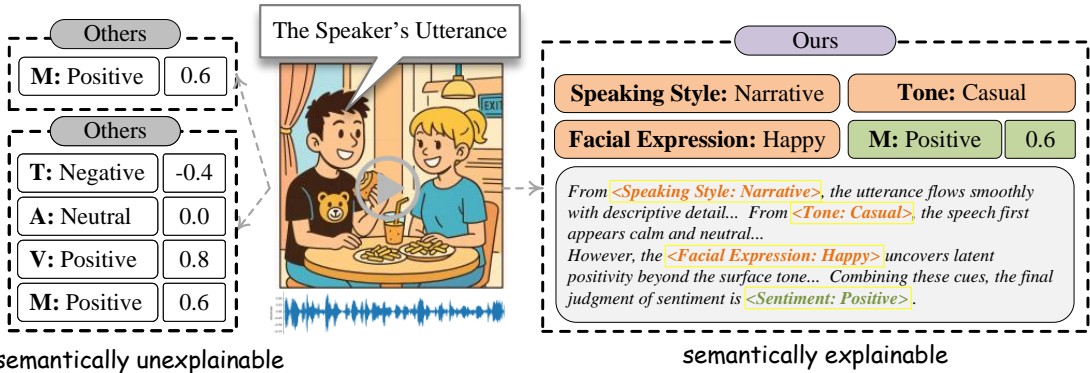

Figure 1: Comparison between conventional datasets and ours. Others provide only modality-level scores (semantically unexplainable), while ours adds concept-level annotations and reasoning chains, making sentiment judgments semantically explainable.

To advance research on explainable multimodal sentiment analysis and enable supervision of the reasoning process, we introduce the first Chinese multimodal dataset, CH-CEMS, with multi-concept annotations for explainable multimodal sentiment analysis. Prior datasets provide only sentiment polarity and intensity annotations of multi- or uni-modalities, without any reasoning cues. To fill in this blank, we further provide annotations of three semantic concepts (speaking style, tone of voice, and facial expression) and two-stage, post-processed chain-of-thought (CoT) traces as the reasoning process. A comparison with prior datasets is shown in Figure 1. Building on this, we propose an explainable multimodal sentiment analysis framework for MLLMs trained with reinforcement learning. By incorporating concept-level supervision as verifiable rewards, our method explicitly models multimodal semantic relations, guiding the model to reason about speakers' sentiment through verifiable concepts and their interactions, and to produce final sentiment judgments grounded in traceable multimodal evidence. We further establish benchmark for both a regression task using five state-of-the-art, feature-based multimodal methods and a generative classification task with mainstream MLLMs. Specifically, we benchmark three closed-source and four open-source MLLMs under zero-shot inference and supervised fine-tuning (SFT) settings. The results reveal a substantial accuracy gap of approximately 2-12% for three-class sentiment analysis between feature-based methods and MLLMs under the generative classification paradigm in the zero-shot setting. While conventional post-training methods improve MLLMs' performance, our approach not only achieves state-of-the-art results on CH-CEMS, surpassing strong closed-source MLLMs such as GPT-5 and Gemini 2.5-Pro, but also produces sentiment predictions with an explainable reasoning process, demonstrating that concept-level supervision within this framework facilitates more faithful and explainable reasoning trajectories and enhances overall performance.

**Contributions.** (1) This paper presents CH-CEMS, the first Chinese dataset for explainable multimodal sentiment analysis. It contains 3,715 high-quality curated samples with not only sentiment polarity and intensity annotations but also reasoning process and cues to facilitate high-level semantics understanding. (2) We develop a reinforcement learning framework for MLLMs that implements concept-level supervision as verifiable rewards, the first successful attempt to leverage reasoning cues for multimodal sentiment analysis. (3) We build a comprehensive benchmark on CH-CEMS. Extensive experiments show that our method achieves state-of-the-art results with substantial improvements over state-of-the-art MLLMs and GPT-5, demonstrating the effectiveness of reasoning concepts for analyzing multimodal sentiment. The CH-CEMS dataset provides a new resource in this area, offering a solid basis in further research.

## 2 RELATED WORK

### 2.1 BENCHMARK DATASETS

**Multimodal Sentiment Analysis Benchmark Dataset.** Early work on multimodal sentiment analysis (MSA) was enabled by video–text–audio benchmark datasets such as CMU-MOSI (Zadeh et al., 2016), CMU-MOSEI (Zadeh et al., 2018), and ICT-MMMO (Wöllmer et al., 2013). These corpora catalyzed research on unimodal representation learning and multimodal fusion, and they remain widely used for training and evaluation. Recent resources enrich supervision beyond utterance-level sentiment. For example, CH-SIMS (Yu et al., 2020) provides independent unimodal annotations in addition to multimodal sentiment (Yu et al., 2020), and CH-SIMS v2.0 (Yu et al., 2023) expands the corpus with a greater focus on non-verbal cues. CMU-MOSEAS extends MSA to multiple non-English languages and incorporates additional emotion and attribute labels (Zadeh et al., 2020). These datasets collectively broaden the scope of MSA and provide finer-grained supervision. Nevertheless, their additional annotations remain largely at the feature or physical signal level, without modeling higher-level semantic concepts that can serve as explicit reasoning clues.

**Explainable Affective Computing Benchmark Datasets.** Recently, several datasets have advanced explainable affective computing. EMER curates multimodal emotion–reasoning pairs (Liu et al., 2023a) and MERR provides a multimodal emotion description and reasoning benchmark (Zhang et al., 2024). PanoSent introduces a multimodal conversational ABSA benchmark for panoptic sextuple extraction and sentiment flipping with causal rationales (Luo et al., 2024). However, to the best of our knowledge, these resources primarily target emotion recognition or aspect-based sentiment analysis rather than user-centric sentiment reasoning. To fill this gap, we present *CH-CEMS*, a concept-centric, reasoning-ready dataset tailored for explainable multimodal sentiment analysis. A comparison between CH-CEMS and other benchmark MSA datasets is presented in Table 1.

### 2.2 MULTIMODAL SENTIMENT ANALYSIS

Multimodal sentiment analysis has advanced through feature-based methods and pre-trained model approaches. These methods can be broadly categorized into early fusion, late fusion, and end-to-end sequence modeling within conventional architectures, as well as methods leveraging pre-trained and large-scale models for more complex MSA tasks.

**Feature-based Methods.** Feature-based methods typically rely on early and late fusion strategies. Early fusion methods, such as MISA (Hazarika et al., 2020), combine modality-specific and modality-invariant features to enhance multimodal affective state prediction. Building on this, DLF (Wang et al., 2025a) introduces a disentangled language-focused fusion framework to improve language-targeted feature extraction. Similarly, MAG-BERT (Rahman et al., 2020a) applies a Multimodal Adaptation Gate to BERT, improving multimodal fusion at the feature level. In contrast, late fusion methods independently predict sentiment for each modality before combining them at the decision level, enhancing robustness. For example, MCIS (Yang et al., 2024a) utilizes counterfactual reasoning to mitigate biases and make robust predictions without additional training. Models such as MulT (Tsai et al., 2019) use an end-to-end approach, employing a multimodal transformer with cross-modal attention to model interactions between modalities without explicit alignment. ALMT (Zhang et al., 2023) enhances model robustness under noisy and imbalanced conditions by guiding multimodal interactions through language.

**Pre-trained Models for MSA.** The rise of pre-trained models has shifted focus from handcrafted feature extraction to leveraging large pre-trained language and vision models. For example, VLP2MSA (Yi et al., 2024) integrates CLIP and TimeSformer to enhance textual prompts from video content for sentiment analysis. Other models, such as those by Yu et al. (Yu et al., 2022) and Yang et al. (Yang et al., 2023), fine-tune pre-trained language models using prompts to improve multimodal alignment for sentiment tasks.

The use of large-scale models has further propelled the performance of MSA. Wang et al. (2024) utilizes large language models to generate contextual information for sentiment analysis, while Feng et al. (2024) integrates vision-language models to reduce noise from images. Shangguan et al. (2025) propose a chain-of-thought reasoning distillation method for large models under resource constraints. Several large-scale models, such as the GPT series and mainstream open-source MLLMs, have been evaluated for zero-shot reasoning, supervised fine-tuning, and instruction tuning in MSA tasks (Song, 2024; Luo et al., 2025; Zhang et al., 2025b), demonstrating their enhanced cognitive capabilities in both multimodal and multilingual contexts. Furthermore, reinforcement learning-based methods for explainable affective computing have emerged as a growing field. R1-Omni (Zhao et al., 2025a) represents a significant step toward explainable emotion recognition. However, MLLM-based EMSA methods remain unexplored due to resource limitations.

Overall, the trend in MSA research reflects a shift from feature-level fusion to methods leveraging pre-trained and large language models, enhancing the performance of multimodal sentiment analysis. To further support the development of explainable multimodal sentiment analysis (EMSA), we introduce the benchmark dataset, CH-CEMS.

## 3 THE CH-CEMS BENCHMARK DATASET

Table 1: Multimodal and Textual Datasets Comparison. ✓indicates availability of the attribute, ✗ indicates absence.

| Dataset | Samples | Speakers | Modalities | Task | Addtional Annotation | Languages | Duration | Domains |
|---|---|---|---|---|---|---|---|---|
| YouTube | 300 | 50 | l, v, a | SP, SI | ✗ | EN | 00:29 | diverse |
| MOUD | 400 | 101 | l, v, a | SP, SI | ✗ | ES | 00:59 | review |
| ICT-MMMO | 340 | 200 | l, v, a | SP, SI | ✗ | EN | 13:58 | movie |
| MOSI | 2,199 | 98 | l, v, a | SP, SI | Subjectivity | EN | 02:36 | diverse |
| MOSEI | 23,453 | 1,000 | l, v, a | SP, SI | ✗ | EN | 65:53 | diverse |
| CH-SIMS | 2281 | 474 | l, v, a | SP, SI | unimodal sentiment | ZH | 02:19 | adverts |
| CMU-MOSEAS | 40,000 | 1,645 | l, v, a | SP, SI | Attributes, Subjectivity | Diverse | 68:49 | diverse |
| CH-SIMS v2.0 | 14563 | – | l, v, a | SP, SI | unimodal sentiment | ZH | – | diverse |
| EMER | 332 | – | l, v, a | SP, SI, EC | Visual Clues, CoT | ZH | – | adverts |
| MERR | 33,105 | – | l, v, a | EC | Attributes,Description | ZH | – | adverts |
| **CH-CEMS (ours)** | **3715** | **1866** | **l, v, a** | **SP, SI** | **Concept, CoT** | **ZH** | **07:54** | **diverse** |

### 3.1 DATA COLLECTION

To approximate real-world scenarios while maintaining diversity, we collect raw videos from six scenarios: vox pops, variety shows, melodramas, formal interviews, vlogs, and science-broadcasting programs. We retain the source resolution, up to 1080p and mostly 720p, during the collection process and use an editing pipeline to segment videos into clips of appropriate length to support utterance-level analysis and predominant on-screen presence of the speaker. Compared with unimodal datasets and multimodal datasets built under earlier conventions, we place greater emphasis on fine-grained cross-modal signals to provide additional supervision. Since these signals depend on reliable audio-visual evidence, we specify three principles for clip acquisition and curation. First, we select clips that contain a complete utterance and are predominantly in Mandarin. Second, we constrain clip length to a proper range and require that the speaker exhibits a consistent sentiment without abrupt shifts. Third, to support multimodal reasoning, the speaker's face is visible for most of the clip and the speech is intelligible without dominant background noise.

### 3.2 DATA ANNOTATION

**Concept Definition.** Additional annotations in prior datasets primarily target unimodal features or sentiment labels to support modality-specific representation learning in late-fusion settings and representation learning across modalities in early-fusion settings. In the generative reasoning paradigm, feature representations are largely determined by large-scale pre-training in MLLMs. To strengthen reasoning during post-training, richer semantic annotations are required as explainable cues. Thus We accordingly annotate three concepts: speaking style {Informational Interaction, General Narrative Exposition} (Biber, 1989), facial expression {happy, sad, surprised, neutral, disgusted, angry}, and tone {casual, happy, sad, wistful, neutral, surprised, disgusted, angry, excited, intimate, nervous, curious, authoritative,

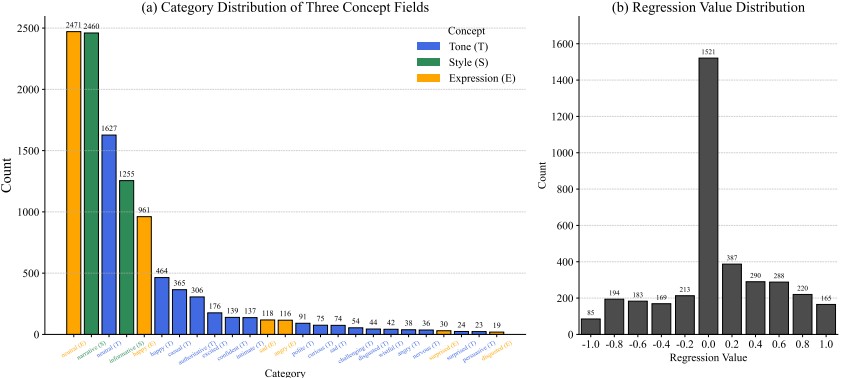

Table 2: Statistics for CH-CEMS

| Metric | Value |
| --- | --- |
| Average utterance length | 34.98 |
| Max/min utterance length | 156/2 |
| Average clip duration(s) | 7.66 |
| Max/min clip duration(s) | 36.46/0.97 |
| Male/Female | 1734/1981 |
| kappa_Tone | 0.54 |
| kappa_Expression | 0.54 |
| kappa_Speaking Style | 0.48 |
| kappa_sentiment | 0.46 |

Figure 2: Distribution of three concept categories (tone, style, expression) and sentiment regression values.

confident, polite, persuasive, challenging, disappointed, indifferent, expectant, sarcastic}. Details of the concepts are illustrated in Appendix B.

**Annotation Process.** The annotation procedure was carried out by a team of five annotators. Each multimodal segment was labeled with three semantic concepts and sentiment label. For concepts, annotators independently assigned labels, and the final decision was determined by a 5-out-of-3 voting scheme. If fewer than three annotators agreed, the instance was sent back for re-annotation until consensus was reached. For sentiment, annotators assigned one of three categories {positive, neutral, negative}, which were mapped to $\{1, 0, -1\}$ respectively. The average score across annotators was then taken as the sentiment value, providing both a categorical polarity and a intensity score for regression-based evaluation. To align with the paradigm of sentiment classification tasks, which are better suited for large language models, we adopted a mapping approach from (Yu et al., 2020; Liu et al., 2022) to map regression values to three class (negative, neutral, positive) or five class (strong negative, weak negative, neutral, weak positive, strong positive) sentiment labels. The full workflow and more details are illustrated in Appendix A (see Figure A)

**Annotation Results.** We ultimately collected 3,715 high-quality clips to build CH-CEMS. For each sample, we provide three concept categories, sentiment annotations, and a reasoning process, as described in Section 4.1. The dataset contains six scenarios and 1,866 speakers, with three interaction types: single-person, multi-person (no interaction), and multi-person (interactive). The dataset's label distributions vary across scenarios and interaction types, which in turn affect the performance of multimodal methods. We analyze these effects in Appendices C and H.1. Neutral sentiment accounts for the largest share (40.1%), followed by positive (36.3%) and negative (22.7%), consistent with the dataset's predominantly real-world sources. Additionally, the category distributions for the three concepts and the sentiment regression values exhibit a long-tailed pattern, as shown in Figure 2. To assess annotation quality, we compute Fleiss' kappa (McHugh, 2012) for the three concepts and for sentiment, and the statistics are reported in Table 2.

### 3.3 BASELINES

We establish benchmark on two tracks: (i) conventional feature-based MSA methods under a regression objective, and (ii) multimodal large language models (MLLMs) under a generative classification objective.

**Traditional regression baselines.** We evaluate five Recent state-of-the-art feature-based MSA methods under the regression paradigm commonly adopted in prior work, and follow the same data splits and evaluation procedures as on CH-SIMS and MOSEI when reporting results on CH-CEMS (see Figure 3). Specifically, **MuLT** (Tsai et al., 2019) is a multimodal transformer that applies directional pairwise cross-modal attention to capture interactions across time; **BERT-MAG** (Rahman et al., 2020b) augments a BERT backbone with a multimodal adaptation gate injected at multiple layers to integrate non-text modalities; **MISA** (Hazarika et al., 2020) jointly learns modality-invariant and modality-specific factors via distributional similarity, orthogonality constraints, reconstruction, and task losses; **DLF** (Liu et al., 2023b) disentangles shared and modality-specific representations with geometric regularization, enhances language features with a Language-Focused Attractor, and performs hierarchical prediction; and **ALMT** (Yu

et al., 2021) is an adaptive language-guided transformer that suppresses irrelevant or conflicting visual/audio signals via an Adaptive Hyper-modality Learning module and fuses the resulting hyper-modality representations.

**Classification baselines.** We evaluate four open-source and three closed-source multimodal large language models in the zero-shot setting. For open-source models, we select the most recent mainstream open-source MLLMs for evaluation, include *MiniCPM-V-4.0* (Yao et al., 2024), *MiniCPM-o-2.6* (Yao et al., 2024), *Qwen2.5-VL-7B* (Bai et al., 2025), and *Qwen2.5-Omni-7B* (Xu et al., 2025). We further optimize *Qwen2.5-Omni-7B* with our concept-guided GRPO. The closed-source baselines are *GPT-4o*, *GPT-5*, and *Gemini 2.5-Pro*. To enable direct comparison with feature-based regression baselines, we map their continuous outputs to three-class or five-class sentiment labels using the same dataset-level mapping and we evaluate all systems with the same classification metrics. Unless otherwise noted, we use a common prompt template, tag schema, input modalities, and decoding settings across models. Results are reported in Table 4 and the significance analysis is illustrated in Appendix G.

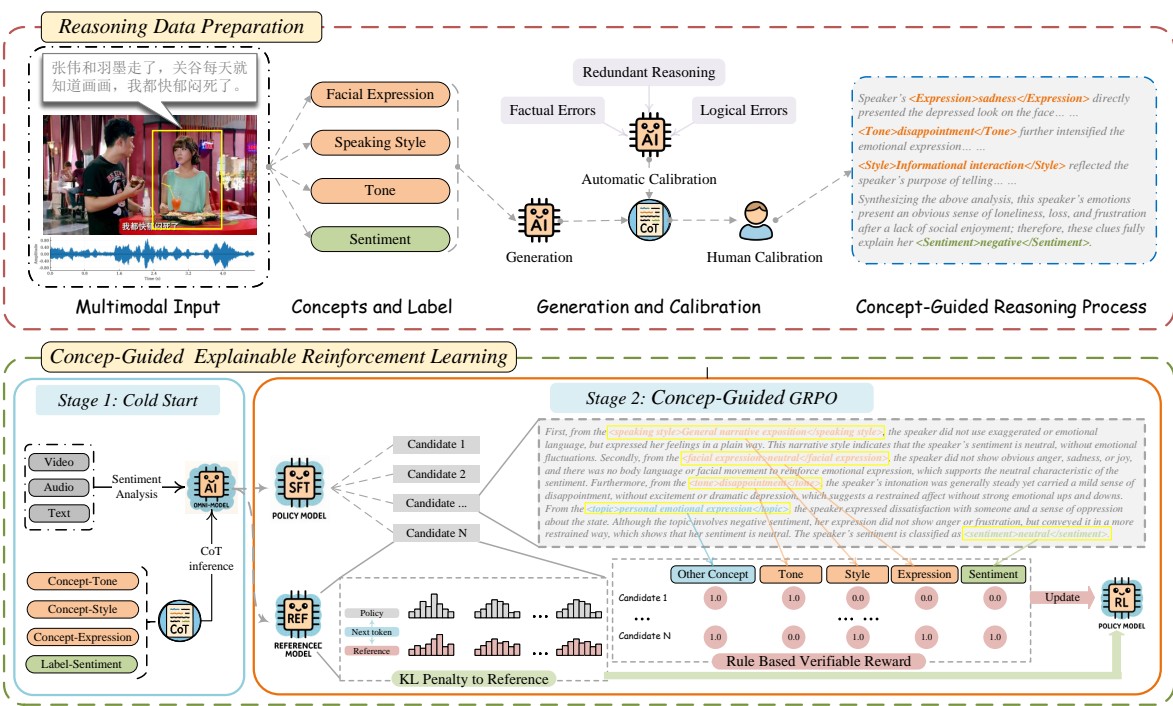

Figure 3: Overview of the Concept-Guided Explainable Reinforcement Learning Framework

Note: Although the training process used Chinese data, we provide a literal English translation here for clarity. Detailed case-study examples in the original Chinese are shown in Table 6.

## 4 CONCEPT-GUIDED EXPLAINABLE REINFORCEMENT LEARNING

This section present a practice of explainable multimodal sentiment analysis based on CH-CEMS by Concept-Guided Explainable Reinforcement Learning Framework, illustrated in Figure 3.

### 4.1 REASONING DATA CONSTRUCTION

To obtain explainable and verifiable sentiment analysis with a structured chain-of-thought, we cold-start training via supervised fine-tuning. We first build a concept-guided reasoning dataset from CH-CEMS with concept-level annotations. Each sample includes multimodal inputs, concept annotations, and a sentiment label, and is submitted to a closed-source multimodal large language model to generate a stepwise reasoning trace. In our setup, we employ GPT-4o and Gemini 2.5-Pro. The model first identifies and summarizes concept-level cues from the multimodal evidence, then analyzes how these cues relate to one another and contribute to the target sentiment, and finally produces the sentiment judgment. To make the reasoning auditable, the analysis is constrained to three predefined concepts that we

annotate: the speaker's facial expression, tone of voice, and speaking style. This ensures that the chain-of-thought is grounded in explainable concepts and follows a consistent template. In the same structured format, we also encourage the model to initiate analyses from other summarizable concepts when supported by the evidence. The prompt used in our approach is detailed in Appendix E.1.

By synthesizing evidence from all modalities and concept categories, the reasoning trace explains why a certain sentiment (positive, negative, or neutral) is perceived. However, due to the lack of audio modality in GPT-4o and Gemini 2.5-Pro and hallucination issues during inference, we apply manual calibration. We polish the reasoning process from three perspectives: (1) Correct factual errors: missing modality may influence the model to infer using non-existent or incorrect facts, such as speaker attribution errors. (2) Reduce redundant reasoning: in an effort to follow instructions, the model may generate concepts and analyses that contribute little or repeat similar arguments. (3) Correct logical errors in the reasoning process: some relationships between concepts are incorrect or strained and should be corrected. The upper panel of Figure 3 illustrates an example of the process.

## 4.2 COLD START WITH SUPERVISED FINE-TUNING

In the first stage of training, we cold-start the model by supervised fine-tuning (SFT) on concept-annotated reasoning traces, adapting a pretrained multimodal large language model via LoRA for parameter efficiency. Concretely, LoRA freezes the pretrained weights and learns a low-rank update on top: for a frozen matrix $W_0 \in \mathbb{R}^{d \times k}$, we introduce trainable $B \in \mathbb{R}^{d \times r}$ and $A \in \mathbb{R}^{r \times k}$ with $r \ll \min(d, k)$,

$$W = W_0 + \Delta W, \qquad \Delta W = BA, \qquad h = W_0 x + BAx, \tag{1}$$

so that only the low-rank parameters $(A, B)$ are updated while $W_0$ remains fixed.

Under this parameterization, we optimize the model parameters $\theta$ to maximize the likelihood of the annotated reasoning sequence $y$, which is achieved by minimizing the cross-entropy between the model output distribution and the reference explanation:

$$\mathcal{L}_{\text{SFT}}(\theta) = - \sum_{t=1}^{T} \log P_\theta\big(y_t \mid y_{<t}, x\big), \tag{2}$$

where $y_t$ denotes the $t$-th token in the ground-truth reasoning sequence and $P_\theta(y_t \mid y_{<t}, x)$ is the model-assigned probability given the preceding tokens and the input.

This training objective aligns the model's behavior with the concept-annotated traces: the model learns to generate a explanatory reasoning process, with the correct `<Style>`, `<Tone>`, and `<Expression>` tags, culminating in the final `<Sentiment>` label. The supervision covers not only the final decision but the entire reasoning path, encouraging the model to internalize how semantic concepts map to sentiment outcomes. After this supervised fine-tuning (SFT) stage, the model can already produce structured explanations for sentiment, effectively acquiring the reasoning template defined by our annotations. This cold-start initialization is critical for stabilizing the next phase, as it provides the policy model $\pi_\theta$ with a strong prior for concept-grounded reasoning. Consequently, subsequent reinforcement learning refines a sensible initial policy rather than starting from scratch and mitigates reward hacking on format-based rewards.

## 4.3 CONCEPT-GUIDED GRPO

After supervised fine-tuning, we further adapt the model with reinforcement learning to explicitly optimize explanation quality and accuracy. We adopt a Concept-Guided Group Relative Policy Optimization algorithm, a policy-gradient method that refines the model using task-specific reward signals. For each input, the policy $\pi_\theta$ generates $G$ candidate reasoning outputs, which are then evaluated by a composite reward function comprising three components. The format reward $R_{\text{format}}$ ensures structural correctness of the explanation, returning 1 only if all required tags appear in the proper structure and 0 otherwise. The answer accuracy reward $R_{\text{acc}}$ encourages correct sentiment classification, assigning 1 when the sentiment inside the `<Sentiment>` tag matches the ground truth. The concept reward $R_{\text{concept}}$ measures the correctness of predicted semantic concepts, defined as

$$R_{\text{concept}} = \frac{1}{K} \sum_{k=1}^{K} \mathbf{1}\{\hat{c}_k = c_k^{GT}\}, \tag{3}$$

where $c_k^{GT}$ is the ground-truth label for concept $k$ and $\hat{c}_k$ the predicted label. The total reward is

$$R_{\text{total}} = R_{\text{format}} + R_{\text{acc}} + R_{\text{concept}}, \tag{4}$$

with a maximum score when all criteria are satisfied.

During optimization, each sampled output $y_i$ is assigned $r_i = R_{\text{total}}(x, y_i)$, and group-relative advantages are computed as

$$A_i = \frac{r_i - \mu_r}{\sigma_r}, \tag{5}$$

where $\mu_r$ and $\sigma_r$ are the mean and standard deviation of the group rewards. These normalized advantages serve as relative-quality signals, guiding the policy update to increase the likelihood of high-reward generations while suppressing low-reward ones. To stabilize training, we include a KL-regularization term that constrains $\pi_\theta$ from deviating excessively from the SFT-initialized reference policy $\pi_{\text{ref}}$. The overall optimization objective is therefore

$$\max_\theta \; \mathbb{E}_{y \sim \pi_\theta}[R_{\text{total}}(x, y)] \; - \; \beta \, \mathbb{E}_x[\mathrm{KL}(\pi_\theta(\cdot|x) \,\|\, \pi_{\text{ref}}(\cdot|x))]. \tag{6}$$

This procedure aligns the model with verifiable rewards covering structural validity, sentiment accuracy, and semantic consistency, thereby encouraging generations that are not only accurate but also accompanied by faithful and interpretable reasoning traces grounded in multimodal evidence, as illustrated in our framework diagram.

## 5 EXPERIMENTS AND DISCUSSION

### 5.1 IMPLEMENTATION DETAILS

We follow the dataset partition shown in Tabel 6, maintaining the predefined train/validation/test splits for both the 3-class and 5-class settings. For conventional feature-based MSA baselines, we adopt the official implementations and run all experiments on a single Tesla V100 GPU with 32 GB memory. For multimodal large language models, we evaluate zero-shot inference, and we train our reinforcement learning framework on eight H20 GPUs with 96 GB memory each. For conventional baselines and MLLM zero-shot inference, we report results over five random seeds (mean and standard deviation) to assess statistical significance in G. The reinforcement learning stage is implemented with the Swift framework (Zhao et al., 2024) using the GRPO backend, with concept-guided rewards as described in Section 4. We initialize the policy from Qwen2.5-Omni-7B, and train with a learning rate of $1 \times 10^{-6}$, a batch size of 8 per device with gradient accumulation of 4, and 3 epochs. The optimizer is AdamW with weight decay, using `bfloat16` precision, the `flash_attn` implementation, and the ZeRO-2 distributed strategy. We set $\beta = 0.04$ for KL regularization, employ 4 sampled generations per prompt with temperature 1.1 and top-$p$ 0.93, and apply gradient clipping at 0.1. Additional training hyperparameters to support reproducibility are available in our repository.

### 5.2 EVALUATION METRICS

We report results under two complementary evaluation paradigms. For conventional regression-based methods, we adopt the standard metrics in MSA: binary accuracy (Acc2), F1-score, weak accuracy, correlation (Corr), coefficient of determination ($R^2$), and mean absolute error (MAE). These metrics capture both classification-oriented and regression-oriented perspectives, enabling fair comparison with prior work. For MLLMs, we follow the recent trend of casting MSA as a classification problem. In this setting, we report overall accuracy (ACC), weighted F1 (WF1), weighted precision (WP), recall (R), and precision (P), under both the 3-class and 5-class settings.

Table 3: Results for feature-based methods on regression task.

| Models | Acc2 (↑) | F1_score (↑) | Acc2_weak (↑) | Corr (↑) | MAE (↓) |
|---|---|---|---|---|---|
| MAG-BERT | 61.67/73.13 | 61.78/75.02 | 50.67 | 56.04 | 28.37 |
| MISA | 65.19/69.34 | 65.73/71.78 | 55.45 | 56.70 | 28.50 |
| MulT | 62.53/71.74 | 62.92/73.89 | 52.18 | 56.00 | 28.47 |
| ALMT | 62.96/72.65 | 63.29/74.70 | 52.37 | 56.09 | 28.95 |
| DLF | 63.61/71.09 | 63.94/73.23 | 53.72 | 56.14 | 28.54 |

### 5.3 RESULTS ON CH-CEMS

**Regression track.** On CH-CEMS, prior feature-based MSA models trained under the regression paradigm exhibit modest absolute scores with no single method dominating all metrics and performance varies across systems. For

Table 4: Classification results on two tasks (3-class and 5-class).

| Models | 3-class | | | | | | 5-class | | | | | |
|---|---|---|---|---|---|---|---|---|---|---|---|---|
| | ACC | WF1 | WP | F1 | R | P | ACC | WF1 | WP | F1 | R | P |
| MAG-BERT | 55.96 | 55.29 | 56.04 | 56.34 | 57.55 | 56.47 | 42.77 | 39.90 | 43.94 | 31.51 | 34.44 | 39.06 |
| MISA | 57.79 | 57.18 | 58.50 | 57.95 | 60.12 | 57.77 | 43.72 | 42.80 | 47.90 | 36.03 | 37.12 | 43.88 |
| MulT | 56.42 | 55.77 | 56.68 | 56.75 | 58.49 | 56.56 | 42.61 | 40.84 | 46.01 | 33.20 | 35.10 | 42.52 |
| ALMT | 55.59 | 54.85 | 55.47 | 55.95 | 57.62 | 55.58 | 41.94 | 41.15 | 44.68 | 36.10 | 36.48 | 41.78 |
| DLF | 57.55 | 57.12 | 58.08 | 58.10 | 59.71 | 58.06 | 44.09 | 42.44 | 45.92 | 35.11 | 36.62 | 42.25 |
| MiniCPM-V-4.0 (ZS) | 52.68 | 46.31 | 56.09 | 49.37 | 56.25 | 56.14 | 28.42 | 22.80 | 28.23 | 18.98 | 35.48 | 17.85 |
| MiniCPM-o-2.6 (ZS) | 62.85 | 62.76 | 64.75 | 62.93 | 62.20 | 65.54 | 40.89 | 41.55 | 44.24 | 39.13 | 41.95 | 38.20 |
| Qwen2.5-VL-7B (ZS) | 61.91 | 61.91 | 70.69 | 61.08 | 65.58 | 65.58 | 31.22 | 31.22 | 48.74 | 31.35 | 38.82 | 38.82 |
| Qwen2.5-Omni-7B (ZS) | 64.98 | 64.89 | 66.48 | 64.61 | 63.12 | 68.38 | 46.92 | 46.65 | 50.78 | 42.04 | 45.67 | 44.88 |
| GPT-4o (ZS) | 67.43 | 67.43 | 67.48 | 68.29 | 68.08 | 68.54 | 44.01 | 44.32 | 48.99 | 41.43 | 45.59 | 43.30 |
| Gemini2.5-Pro (ZS) | 68.51 | 66.85 | 71.66 | 68.50 | 71.07 | 71.45 | 37.28 | 37.03 | 53.17 | 37.87 | 49.24 | 42.39 |
| GPT-5 (ZS) | 70.12 | 69.21 | _71.69_ | 70.45 | 71.86 | _71.94_ | 45.49 | 44.64 | 54.60 | 47.40 | **54.32** | 48.66 |
| MiniCPM-V-4.0 (SFT) | 62.45 | 62.42 | 63.61 | 62.26 | 60.92 | 65.23 | 45.49 | 38.38 | 43.83 | 27.95 | 32.71 | 34.09 |
| MiniCPM-o-2.6 (SFT) | 65.01 | 64.97 | 65.50 | 65.63 | 65.83 | 65.95 | 49.53 | 49.54 | 50.16 | 32.70 | 32.93 | 33.22 |
| Qwen2.5-VL-7B (SFT) | _71.60_ | _71.53_ | 71.62 | _71.66_ | _72.13_ | 71.34 | _55.59_ | _54.79_ | **55.89** | _48.22_ | 46.69 | **53.43** |
| Qwen2.5-Omni-7B(SFT) | 70.12 | 70.14 | 70.38 | 70.42 | 70.14 | 70.91 | 52.89 | 50.12 | 51.96 | 47.79 | 50.48 | 50.93 |
| Qwen2.5-Omni-7B (GRPO) | 69.18 | 69.44 | 70.09 | 69.07 | 69.21 | 69.56 | 51.68 | 46.31 | 48.27 | 47.30 | 50.91 | _52.05_ |
| CD-GRPO (ours) | **73.62** | **73.61** | **75.56** | **74.06** | **74.77** | **75.03** | **56.39** | **54.86** | _55.60_ | **51.75** | _53.76_ | 51.83 |

example, some models favor correlation and error while others are relatively stronger on Acc2 or F1, indicating the increased difficulty and distribution shift in CH-CEMS. Representative numbers are reported in Table 3.

**Classification track.** In the 3-class setting, a clear gap emerges: small feature-based models trail strong MLLM baselines by about 10 percentage points in accuracy. Among off-the-shelf MLLMs, GPT-5 is the strongest closed-source baseline. Building on an open-source backbone, our concept-guided post-training on Qwen2.5-Omni-7B achieves 73.62% accuracy on the three-class sentiment task (Table 4), exceeding GPT-5 by 3.50 points and open-source SFT baselines by 2.0–10.0 points, with consistent gains in WF1, WP, F1, recall, and precision. In the 5-class setting, the gap narrows: several traditional baselines match or exceed closed-source models on some metrics, and GPT-5 attains the strongest accuracy among evaluated MLLMs under zero-shot inference. CD-GRPO achieves performance comparable to Qwen2.5-VL-7B under SFT and surpasses most other methods, attaining top scores on three metrics. Unlike label-only predictors, it also outputs concept-grounded, verifiable reasoning traces which is demonstrated in Figure 6. Overall, large models excel at polarity recognition (3-class), while fine-grained intensity (5-class) remains challenging. The performance of CD-GRPO demonstrates the effectiveness of concept-level semantic cues from CH-CEMS under a reinforcement learning paradigm and offers a solid basis for further research on explainable multimodal sentiment analysis.

## 6 CONCLUSIONS

Motivated by the growing interest in explainable affective computing and the fact that while explainable datasets are plentiful for emotion recognition they remain scarce for sentiment analysis, we introduce CH-CEMS, a multimodal sentiment dataset with concept-level annotations and accompanying reasoning traces towards EMSA. Building on CH-CEMS, we propose a training pipeline that cold-starts the model via supervised fine-tuning on structured explanations and then applies rule-based reinforcement learning with concept-level rewards, providing verifiable supervision over the reasoning process. To support research on CH-CEMS, we benchmark both feature-based regression baselines and generative classification with four open- and three closed-source multimodal large language models. On our benchmark, MLLMs exhibit strong overall performance, and our concept-guided GRPO on Qwen2.5-Omni-7B surpasses strong closed-source baselines such as GPT-5 while producing structured reasoning traces under a unified tag schema, highlighting the effectiveness of semantic cues and a promising direction for explainable multimodal sentiment analysis in the generative paradigm.

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

## A DATASET CONSTRUCTION PIPELINE

### A.1 DATA SOURCE

Our dataset was collected from Bilibili and comprises unedited video clips drawn from vox pops, variety shows, melodramas, formal interviews, vlogs, and other real-world scenarios. These videos capture rich, multimodal emotional expressions in natural contexts, offering high authenticity and diversity. More than 97% of the videos have a resolution of at least 720p, ensuring visual clarity and providing high-quality audiovisual inputs for fine-grained sentiment analysis.

### A.2 DATA PROCESSING

Due to variations in shooting angles, distances, and lighting conditions in the original videos, the environments in which the speakers are situated exhibit considerable complexity. In addition, the audio may contain minor noise, such as background music or overlapping monologues, and the speakers' speech rate and tone may also vary. To construct a high-quality multimodal sentiment analysis dataset, we designed a systematic data processing pipeline consisting of two key steps: video filtering and speech utteranceion.

**Video Clip Filtering.** All video clips were required to retain their original resolution and be stored in MP4 format. Using dedicated cropping tools, the original videos were trimmed into segments no longer than two minutes, which were subsequently divided into shorter utterances. A semantic filtering procedure was then applied to automatically remove clips lacking substantive semantic content.

Building on this step, we introduced a manual screening mechanism to further enhance data quality. The criteria for this second stage were as follows: for the visual modality, the video must clearly display the speaker's face, and the speaker must remain in the frame for more than half of the segment's duration to ensure effective visual information capture; for the auditory modality, the speaker's voice must be clear, and the intensity of background music or noise must not interfere with the speech signal, thereby ensuring the recognizability and consistency of the audio. Two professionally trained annotators participated in this stage, performing strict screening from the perspective of multimodal reliability, which ultimately yielded high-quality sample instances.

**Text utteranceion.** To obtain textual data that accurately aligns with the audio-visual content, we employed iFlytek's closed-source speech recognition API for initial utteranceion. All transcribed texts were then manually proofread to correct typos, sentence segmentation, and punctuation errors. Personally identifiable information (such as names) was anonymized. This process substantially improved the accuracy, consistency, and security of the textual data, providing a reliable linguistic foundation for multimodal sentiment analysis.

## A.3 DATA ANNOTATION

In the data annotation phase, we independently developed an efficient multimodal sentiment annotation platform and established a unified database for both concept and sentiment labels, thereby significantly enhancing annotation efficiency and data quality. The platform was designed to enable seamless interaction between annotators and multimodal data, featuring an intuitive and user-friendly interface, as illustrated in Figure 4.

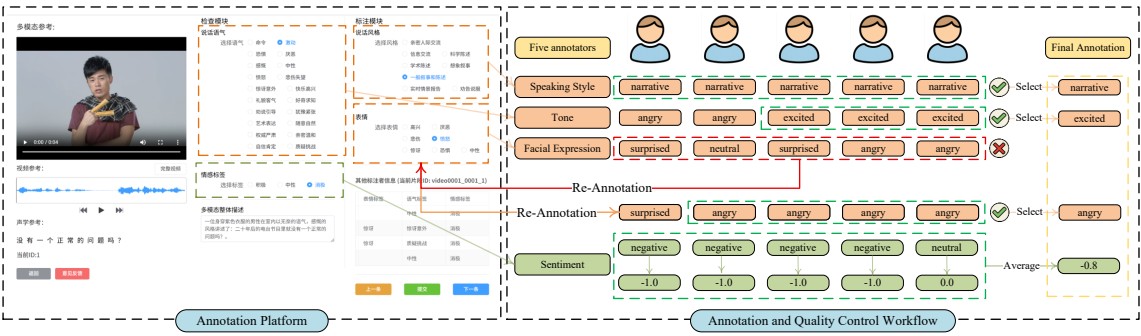

Figure 4: Workflow for dataset annotation: item selection, multi-annotator labeling, quality assurance review, re-annotation, and label finalization.

The platform supports two core annotation tasks: concept labeling and sentiment labeling. Concept labels span three dimensions: the speaker's tone, speaking style, and facial expressions. By comprehensively watching videos, listening to audio, and reading text, annotators extract information from the visual, auditory, and textual modalities to complete concept annotations, and subsequently determine the overall sentiment label through multimodal information fusion. The annotation process is accomplished through simple point-and-click interactions, which greatly reduce operational complexity and learning costs, while significantly accelerating the annotation workflow. All annotation results are stored in the database in real time for subsequent statistical analysis. In addition, descriptive text was generated for each multimodal sample, providing a valuable resource for researchers conducting further studies in multimodal sentiment analysis.

We also present a representative case of the constructed CH-CEMS. A total of five professional annotators participated in labeling both concept and sentiment categories. The platform's user-friendly interface allows annotators to conveniently compare video and text content, thereby ensuring annotation accuracy and consistency.

During the concept labeling stage, we introduced a "five-choose-three" voting mechanism, requiring that each semantic concept be selected by at least three annotators to be confirmed. To address the sparsity of samples in certain speaking-style categories, we merged and optimized them: "intimate interpersonal communication" and "information exchange" were unified into "information exchange," while other styles were grouped into "narrative." Labels that failed to reach

consensus were returned for re-discussion and re-annotation, ensuring both high quality and consistency of the concept labels.

Sentiment labeling was built upon the foundation of concept annotations. We adopted a polarity-based sentiment quantification scheme, categorizing sentiments into positive (1.0), neutral (0.0), and negative (-1.0). This approach is consistent with the sentiment labeling methods employed in CH-SIMS (Yu et al., 2020), CH-SIMS v2.0 (Yu et al., 2023), CMU-MOSI (Zadeh et al., 2016), and CMU-MOSEI (Zadeh et al., 2018). By summing and averaging the scores assigned by each annotator, we derived continuous sentiment regression values, thereby providing a fine-grained depiction of the speaker's sentiment state.

This systematic annotation process not only guarantees the reliability and consistency of the annotated data—laying a solid foundation for building robust, high-performance multimodal sentiment analysis models—but also facilitates efficient data collection and serves as a valuable tool for exploring the complex relationships between semantic concepts and sentiments.

## B  CONCEPT DEFINITIONS

**Speaking Style label taxonomy.**   We adopt Biber's data-driven register framework because it defines speaking style by statistically induced functional dimensions rather than preset genres. Concretely, large balanced corpora are annotated with dozens of lexical–grammatical features; factor analysis over feature co-occurrence yields five continuous dimensions (Involved vs. Informational production, Narrative vs. non-narrative concerns, Elaborated vs. situation-dependent reference, Overt expression of persuasion, Abstract vs. non-abstract style). Clustering texts in this space gives eight empirically grounded prototypes as illustrated in Table 5. This framework is attractive for concept annotation because it: (i) ties labels to measurable linguistic cues, (ii) provides interpretable axes and prototype labels, and (iii) reduces subjective, task-specific labeling (Biber, 1988; 1989).

In our corpus, many prototype classes are too sparse for reliable supervision. We therefore focus our speaking style annotation on two frequent and complementary prototypes: Informational interaction and General narrative exposition. These labels capture the dominant interactional–informational and mixed narrative–expository styles in our data, while keeping annotation consistent with the underlying dimensional view of style.

Table 5: Eight prototype text types in Biber's typology and concise definitions

| ID | Prototype name | Concise definition and explanation |
|----|----------------|-----------------------------------|
| 1 | Intimate interpersonal interaction | Concerned primarily with the immediate interpersonal interaction. |
| 2 | Informational interaction | Has a primary informational emphasis. |
| 3 | Scientific exposition | Extremely informational, elaborated in reference, and technical and abstract in style and content. |
| 4 | Learned exposition | Similar to Scientific exposition except that it is markedly less abstract and less technical in style. |
| 5 | Imaginative narrative | A relatively involved text type having a primary narrative focus. |
| 6 | General narrative exposition | A very general text type that combines narrative forms with expository, informational elaboration. |
| 7 | Situated reportage | Reporting events actually in progress. |
| 8 | Involved persuasion | Primarily distinguished by persuasive and argumentative emphases, typically combined with an involved (often interactive) style. |

**Facial expression label taxonomy.**   We annotate six facial expression categories (happy, sad, surprised, neutral, disgusted, and angry) because they align with the widely used basic-emotion taxonomy and its operationalization via the Facial Action Coding System (FACS), which ties labels to observable action units and supports reproducible coding (Ekman, 1992; Ekman et al., 2002). The same categorical set (with a neutral baseline) is standard in mainstream FER benchmarks, enabling direct reuse of models and fair comparison across datasets (Lucey et al., 2010). Practically, we replace fear with neutral to improve annotation reliability and class balance for limited fear sample.

**Tone label taxonomy.**   We ground our tone labels in two complementary traditions. First, affective tone is anchored in classic emotion theory: discrete emotions (happy, sad, angry, disgusted, surprised) and a neutral baseline, together

with valence–arousal nuances (excited, nervous, wistful) that are well captured by dimensional models and known to surface in prosody and voice quality (Ekman, 1992; Russell, 1980; Scherer, 2003). Second, interpersonal tone (casual, intimate, polite, persuasive, challenging, authoritative, confident, curious) draws on pragmatics of facework and stance as well as register/style research, linking these labels to recognizable linguistic resources (e.g., mitigation, commitment displays, formality cues) (Brown & Levinson, 1987; Biber, 1988). Practically, we open-coded candidate tone tags on a seed set and consolidated them into the above inventory under the constraints that (i) core valence–arousal quadrants are covered, and (ii) key interpersonal functions in dialogue are represented, yielding a compact, theory-aligned taxonomy for downstream modeling and evaluation.

## C  DETAILED INFORMATION OF DATASET

We collected raw videos from various scenarios, including vox pops, variety shows, melodramas, formal interviews, vlogs, and science-broadcasting programs. We observe that the sentiment distribution varies across scenarios, indicating a scenario-level distribution shift. Besides, prior multimodal sentiment datasets often restrict clips to a single on-screen speaker to avoid confounds from additional people. Accordingly, we annotate the number of on-screen participants and their interaction relations. Sentiment distributions conditioned on both factors (scenario and speaker count/interaction) are shown in Figure 5. For training and evaluation on CH-CEMS, we split the dataset into train/dev/test sets using a 6:2:2 ratio and perform a stratified split to maintain the sentiment-label distribution across splits. The resulting distributions are reported in Table 6

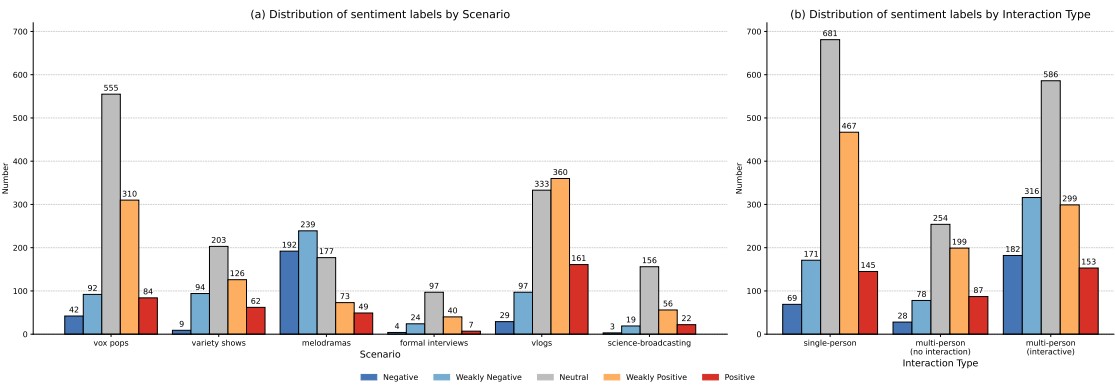

Figure 5: Distribution of five-level sentiment labels (negative, weakly negative, neutral, weakly positive, positive) across (a) six scenarios (vox pops, variety shows, melodramas, formal interviews, vlogs, and science-broadcasting programs) and (b) three interaction types (single-person, multi-person without interaction, and multi-person with interaction)

| Item | Total | 5-class | | | | | 3-class | | |
|---|---|---|---|---|---|---|---|---|---|
| | | NEG | WNEG | NEU | WPOS | POS | NEG | NEU | POS |
| #Train | 2229 | 167 (7.5%) | 339 (15.2%) | 913 (41.0%) | 579 (26.0%) | 231 (10.4%) | 506 (22.7%) | 913 (41.0%) | 810 (36.3%) |
| #Valid | 743 | 56 (7.5%) | 113 (15.2%) | 304 (40.9%) | 193 (26.0%) | 77 (10.4%) | 169 (22.7%) | 304 (40.9%) | 270 (36.3%) |
| #Test | 743 | 56 (7.5%) | 113 (15.2%) | 304 (40.9%) | 193 (26.0%) | 77 (10.4%) | 169 (22.7%) | 304 (40.9%) | 270 (36.3%) |

Table 6: Label distribution of the dataset in training, validation, and test sets. Numbers in parentheses indicate the percentage of each class. For 5-class classification: NEG (Negative), WNEG (Weak Negative), NEU (Neutral), WPOS (Weak Positive), POS (Positive); for 3-class classification: NEG (Negative), NEU (Neutral), POS (Positive).

## D  ADDITIONAL RELATED WORK

### D.1  FEATURE-BASED MSA METHODS

Research on multimodal sentiment analysis (MSA) mainly focuses on two directions: early fusion and late fusion. Early fusion methods aim to capture cross-modal interactions and consistency at the feature level. Among them,

MISA (Hazarika et al., 2020)proposes modality-invariant and modality-specific representations to improve multimodal fusion for affective state prediction. Building on the idea of disentangled representations, DLF (Wang et al., 2025a) proposes a Disentangled-Language-Focused (DLF) multimodal representation learning framework to further enhance language-targeted features. Similarly, MAG-BERT (Rahman et al., 2020a) enhances the multimodal fusion capability of pre-trained models through the Multimodal Adaptation Gate, highlighting the effectiveness of early fusion at the feature level.

Beyond conventional early fusion methods, some works focus on end-to-end modeling of multimodal sequences. For instance, MulT (Tsai et al., 2019) is an end-to-end multimodal Transformer that models interactions and adaptations between multimodal sequences without explicit alignment through directional pairwise cross-modal attention. ALMT (Zhang et al., 2023) presents the Adaptive Language-guided Multimodal Transformer to enhance robustness under noise and modality imbalance, emphasizing the importance of model robustness. Moreover, UniMSE (Hu et al., 2022) leverages joint modeling of MSA and ERC combined with contrastive learning to improve cross-task performance, demonstrating the benefit of exploiting task similarities.

Semi-supervised learning and modality heterogeneity are also actively explored. MC-Teacher utilizes a teacher-student framework augmented with consistency-based pseudo-labels to perform semi-supervised multimodal sentiment analysis, effectively addressing the challenge of limited annotated data. To handle modality heterogeneity, MCL-MCF (Fan et al., 2024) is a progressive multimodal fusion method that alleviates modality heterogeneity at multiple levels through multi-level contrastive learning and tensor convolution fusion, enabling continuous feature-level integration across unimodal, bimodal, and higher-level fused modalities.

In contrast, late fusion combines independent predictions of each modality at the decision level to enhance robustness and flexibility. Finally, MCIS (Yang et al., 2024a) leverages counterfactual reasoning during inference to mitigate biases and achieve robust decision-making without additional training, providing a complementary approach to handling dataset biases in MSA.

## D.2   PRE-TRAINED MODELS FOR MSA

In recent years, with the rapid development of pre-trained models, the research focus of multimodal sentiment analysis has shifted from traditional cross-modal feature interaction to methods based on pre-trained models. For instance, (Yi et al., 2024) combined CLIP and Timesformer architectures to generate discriminative textual prompts from video content, thereby enhancing textual representations and enabling end-to-end sentiment analysis. (Yu et al., 2022) employed prompt phrases to fine-tune frozen language models, obtaining coherent vision-language representations to bridge the semantic gap and reduce parameter dependency. (Yu & Zhang, 2022) integrated visual information into pre-trained language models and utilized prompt tuning to minimize the discrepancy between masked language modeling and sentiment analysis tasks. (Yang et al., 2023) proposed a generative prompt model that constructs multimodal prompts through an encoder-decoder architecture, reducing the reliance on annotated data. (Khan & Fu, 2021) designed a dual-stream pre-trained model that converts images into auxiliary sentences via target-aware transformation and single-pass non-autoregressive generation, thereby injecting multimodal information. (Ling et al., 2022) introduced language-, vision-, and multimodal-specific pre-training tasks within an encoder-decoder framework to facilitate cross-modal alignment. (Ye et al., 2022) proposed a sentiment-aware pre-training framework that captures fine-grained emotional signals from data through cross-modal contrastive learning and additional sentiment objectives.

With the rise of large-scale models, the performance of multimodal sentiment analysis has been further significantly improved. (Wang et al., 2024) leveraged large language models to generate rich contextual information for enhanced sentiment understanding. (Feng et al., 2024) utilized large-scale vision-language models to integrate textual and visual information, mitigating the interference of image noise in sentiment classification. (Xiao et al., 2025) extracted information from visual features and relied on large models to generate emotional causes and impressions to assist analysis. (Shangguan et al., 2025) proposed a multimodal chain-of-thought reasoning distillation method to alleviate model training challenges under resource-constrained conditions. Moreover, several studies have systematically evaluated the performance of large models in multimodal sentiment analysis tasks: (Song, 2024) pioneered the use of large text models and GPT-3.5 Turbo for zero-shot reasoning. (Zhang et al., 2025b) further expanded the evaluation methods to include zero-shot reasoning, instruction tuning, and supervised fine-tuning. (Liu et al., 2025) established extensive benchmarks in multilingual and multimodal scenarios, thoroughly validating the powerful cognitive and reasoning capabilities of large models in this field.

# E  PROMPT USED

## E.1  PROMPT FOR THE CONSTRUCTION OF REASONING PROCESS DATA

During construction of the reasoning dataset, we employ two strong closed-source multimodal models, namely GPT-4o and Gemini 2.5-Pro, to generate chain-of-thought rationales conditioned on multimodal inputs together with the ground-truth concept labels and the sentiment label. Specifically, we use the prompt in Listing 1 to instruct the models to analyze the final sentiment on the basis of the provided concept cues.

Listing 1: Prompt for Sentiment CoT Generation

```
You are an expert in human sentiment analysis. You will receive a video, the
speaker's utterance, some concept clues about the speaker, and the speaker's final
sentiment. I need you to analyze, based on the above, from different concept clues
why the speaker's sentiment is like this.

In addition to the given concept clues, you also need to summarize other concepts
that you consider helpful for analyzing the speaker's sentiment and, based on the
concept clues, perform the analysis. Finally, you need to output your thinking
process.
The given concept clues and the newly discovered concept clues need to be marked
with <concept>...</concept>.

For example: <think>The speaker's facial expression is
<facial_expression>xxx</facial_expression> ... The speaker's speaking style is
<speaking_style>xxx</speaking_style>, combined with semantic analysis it ... The
tone is <tone>xxx</tone>, etc.</think> <sentiment>xxx</sentiment>

The speaker's utterance: '<insert utterance>'.
Concept clues: <speaking_style>xxx</speaking_style>,
<facial_expression>xxx</facial_expression>, <tone>xxx</tone>.
Sentiment label: <sentiment>xxx</sentiment>.
```

After collecting the raw reasoning traces, we manually inspected and corrected them. Because instruction-following behavior can sometimes produce strained explanations, and because the audio modality is occasionally missing and the processing of visual information can be inaccurate, the raw rationales contained many issues. We therefore corrected the data along the three perspectives described in Section 4.1. The correction pipeline proceeds in two stages: we first apply an automatic correction with a closed-source model, using the prompt in Listing 2 to propose edits, and then human annotators verify and amend the outputs.

Listing 2: Prompt for CoT refinement

```
Please correct and refine a chain-of-thought reasoning sample according to the
following requirements:

1) Correct factual errors. Compare all conceptual and factual statements against
the ground-truth label(s). If discrepancies exist, fix them and revise the
reasoning steps accordingly.

2) Remove redundancy. Where the analysis is repetitive, streamline it while
preserving the completeness and coherence of the reasoning.

3) Repair reasoning logic. Where the reasoning contains logical errors or overly
tenuous arguments, revise it to be sound while maintaining the overall reasoning
structure. You must not invent new concepts or add content that is not supported
by the provided information or labels.
```

## E.2  PROMPT FOR METHODS

For the benchmark on CH-CEMS for the classification setting, we follow prior work (Zhang et al., 2025a; Luo et al., 2025; Zhang et al., 2025b) and design standardized zero-shot inference and supervised fine-tuning prompts, such as the five-class sentiment classification task shown in Listing 3, to evaluate both open-source and closed-source models.

Furthermore, we design a new prompt in Listing 4 based on the previous prompt for concept-guided reinforcement learning, as comprehensively detailed in Sections 4.2 and Sections 4.3.

Listing 3: Prompt for Zero-Shot Evaluation and SFT of MLLMs in Five-Class Sentiment Classification

```
You are an expert in human sentiment analysis. You will be given a short video
represented by four images, along with the speaker's utterance and audio. Analyze
the speaker's sentiment based on these inputs.

At the end, output exactly one label from the set: [Positive, Weakly Positive,
Neutral, Weakly Negative, Negative].

The content spoken in the video is: <utterance>
```

Listing 4: Prompt for Explainable Reasoning in Five-Class Sentiment (SFT Cold Start & Concept-Guided GRPO)

```
You are an expert in human sentiment analysis. You will receive a short video
together with the speaker's utterance and audio.
Reason over all concepts you consider useful for inferring the speaker's sentiment.

At the end, choose exactly one label from [Positive, Weakly Positive, Neutral,
Weakly Negative, Negative].
Output your reasoning inside <think></think>, and mark the final sentiment using
<sentiment></sentiment>.

For example: <think>The speaker's facial expression is
<facial_expression>xxx</facial_expression> ... The speaking style is
<speaking_style>xxx</speaking_style> ... The tone is <tone>xxx</tone> ...</think>
<sentiment>xxx</sentiment>

Candidate labels for tone: [<insert tone labels>].
Candidate labels for facial expression: [<insert expression labels>].
Candidate labels for speaking style: [<insert style labels>].
Other useful concepts have no predefined list; analyze them using multimodal cues.

The utterance is: '<insert text>'
```

## F  CASE STUDIES

## G  SIGNIFICANCE ANALYSIS

Table 7: regression.

| Models | Acc2 ($\uparrow$) | F1_score ($\uparrow$) | Acc2_weak ($\uparrow$) | Corr ($\uparrow$) | MAE ($\downarrow$) |
|---|---|---|---|---|---|
| MAG-BERT | 61.67±2.82/73.13±3.75 | 61.78±3.16/75.02±3.31 | 50.67±3.68 | 56.04±1.50 | 28.37±0.44 |
| MISA | 65.19±1.65/69.34±1.59 | 65.73±1.71/71.78±1.42 | 55.45±2.36 | 56.70±0.55 | 28.50±0.43 |
| MulT | 62.53±1.80/71.74±3.23 | 62.92±1.98/73.89±2.86 | 52.18±2.79 | 56.00±0.75 | 28.47±0.48 |
| ALMT | 62.96±1.68/72.65±2.35 | 63.29±1.89/74.70±2.04 | 52.37±2.77 | 56.09±0.21 | 28.95±0.15 |
| DLF | 63.61±3.88/71.09±5.33 | 63.94±4.05/73.23±4.84 | 53.72±6.14 | 56.14±0.93 | 28.54±0.83 |

## H  IMPACT OF DATASET DISTRIBUTION ON RESULTS

### H.1  SCENARIO-WISE LABEL DISTRIBUTION AND MODEL PERFORMANCE

In multimodal sentiment analysis, emotional expression is strongly influenced by the interplay and integration of multiple modalities. Due to the specific encoding conventions and presentation styles of different video types, identical emotional expressions may convey distinct semantic meanings, which substantially increases model discriminative

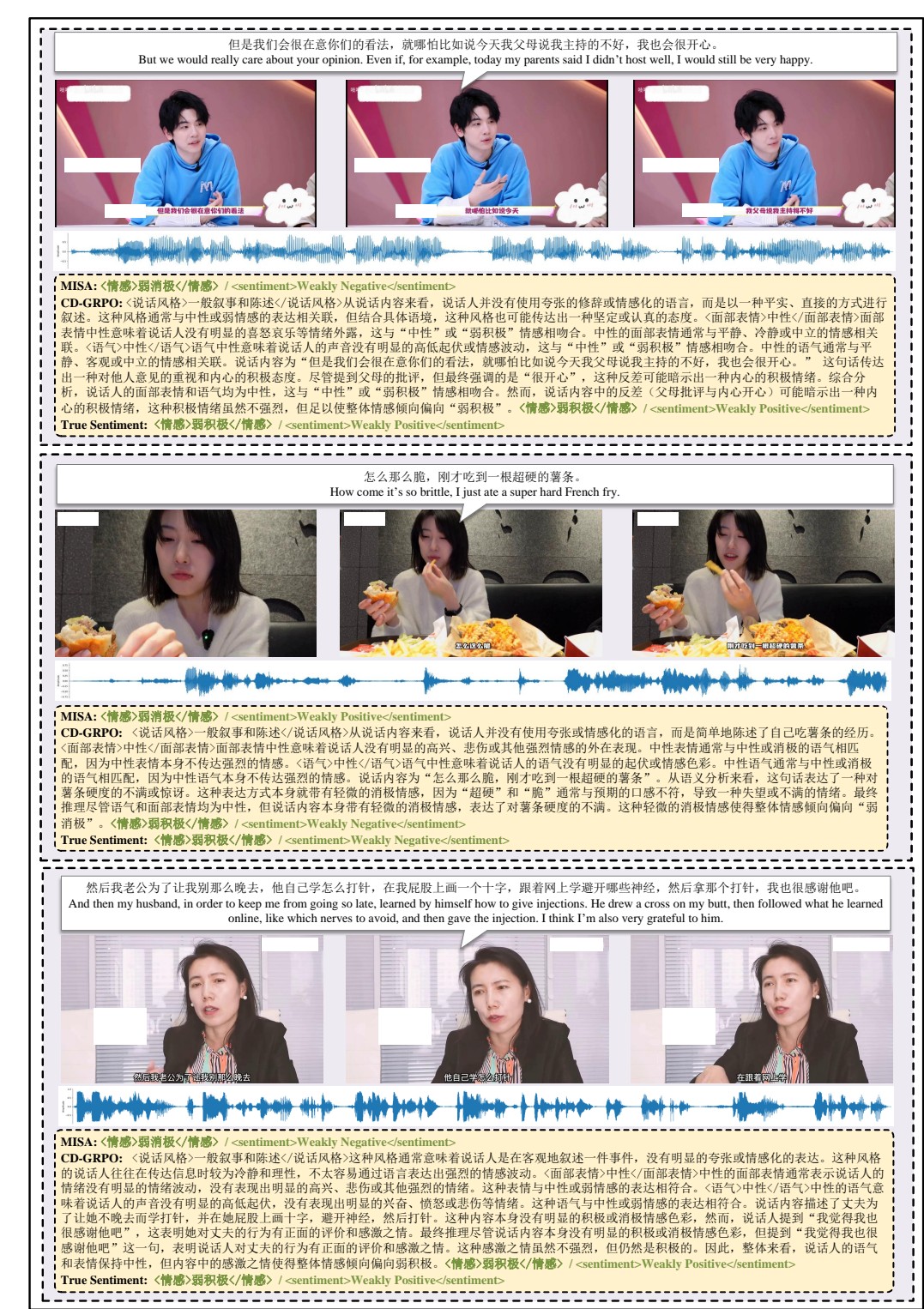

Figure 6: Case study of comparison between MISA and CD-GRPO.

difficulty and induces performance variability. In this section, we present the accuracies of conventional SOTA approaches and multimodal large language models (MLLMs), specifically Qwen2.5-Omni-7B and MiniCPM-o-2.6, for three-class and five-class sentiment recognition across six scenarios: melodramas, vox pops, science broadcasting,

Table 8: 3-class.

| Models | Acc | WF1 | WP | F1 | R | P |
|---|---|---|---|---|---|---|
| MAG-BERT | 55.96±1.03 | 55.29±1.20 | 56.04±0.90 | 56.34±1.23 | 57.55±1.34 | 56.47±1.01 |
| MISA | 57.79±1.23 | 57.18±1.28 | 58.50±1.27 | 57.95±1.18 | 60.12±0.88 | 57.77±1.27 |
| MulT | 56.42±1.35 | 55.77±1.47 | 56.68±1.15 | 56.75±1.49 | 58.49±0.87 | 56.56±1.58 |
| ALMT | 55.59±1.21 | 54.85±1.33 | 55.47±1.34 | 55.95±1.15 | 57.62±1.22 | 55.58±1.06 |
| DLF | 57.55±1.69 | 57.12±1.86 | 58.08±2.13 | 58.10±1.63 | 59.71±1.63 | 58.06±1.85 |
| MiniCPM-V-4.0 | 52.68±0.51 | 46.31±0.69 | 56.09±1.21 | 49.37±0.60 | 56.25±0.52 | 56.14±0.95 |
| Qwen2.5-VL-7B | 61.91±0.00 | 61.91±3.00 | 70.69±4.28 | 61.08±0.00 | 65.58±0.20 | 65.58±0.20 |
| MiniCPM-o-2_6 | 62.85±0.90 | 62.76±0.94 | 64.75±0.87 | 62.93±1.02 | 62.20±1.04 | 65.54±0.95 |
| Qwen2.5-Omni-7B | 64.98±0.43 | 64.89±0.44 | 66.48±0.35 | 64.61±0.50 | 63.12±0.53 | 68.38±0.39 |

Table 9: 5-class.

| Models | Acc | WF1 | WP | F1 | R | P |
|---|---|---|---|---|---|---|
| MAG-BERT | 42.77±0.87 | 39.90±1.32 | 43.94±3.15 | 31.51±3.20 | 34.44±1.73 | 39.06±7.02 |
| MISA | 43.72±1.18 | 42.80±0.97 | 47.90±1.42 | 36.03±0.96 | 37.12±0.77 | 43.88±2.07 |
| MulT | 42.61±1.81 | 40.84±1.59 | 46.01±1.74 | 33.20±1.75 | 35.10±0.99 | 42.52±3.51 |
| ALMT | 41.94±1.21 | 41.15±1.24 | 44.68±1.28 | 36.10±1.18 | 36.48±1.05 | 41.78±1.12 |
| DLF | 44.09±1.62 | 42.44±1.43 | 45.92±1.59 | 35.11±1.93 | 36.62±1.75 | 42.25±1.54 |
| MiniCPM-V-4.0 | 28.42±0.20 | 22.80±0.29 | 28.23±0.30 | 18.98±1.53 | 35.48±3.11 | 17.85±1.44 |
| Qwen2.5-VL-7B | 31.22±0.00 | 31.22±1.31 | 48.74±1.86 | 31.35±0.00 | 38.82±0.43 | 38.82±0.43 |
| MiniCPM-o-2_6 | 40.89±0.84 | 41.55±0.82 | 44.24±0.90 | 39.13±3.55 | 41.95±3.89 | 38.20±3.33 |
| Qwen2.5-Omni-7B | 46.92±0.79 | 46.65±0.84 | 50.78±0.93 | 42.04±2.87 | 45.67±3.24 | 44.88±3.05 |

formal interviews, variety shows, and vlogs. The models were evaluated under random seeds 0–4, and the results are illustrated in Figure 7.

Based on the boxplot results, models perform particularly well in melodramas and vlogs, showing higher median accuracies, narrower interquartile ranges, and more concentrated data distributions, indicating strong robustness in these scenarios. In contrast, in vox pops, although the interquartile ranges are relatively narrow, the high number of outliers suggests limited generalization capability and reduced stability. In science-broadcasting programs, formal interviews, and variety shows, the diversity of content themes and the richness of emotional expressions contribute to greater fluctuations in prediction results, accompanied by markedly expanded box widths, reflecting increased task complexity. Importantly, in variety shows—where filming captures spontaneous reactions from multiple participants—emotional signals are highly heterogeneous and dispersed, resulting in the lowest accuracy among all scenarios. In formal interviews, limited sample sizes lead to the greatest variability in model performance. Across all scenarios, five-class tasks generally achieve lower accuracy than three-class tasks, primarily due to finer-grained categories, sparse sample distributions, ambiguous inter-class boundaries, and higher discriminative demands, which further amplify the overall complexity of sentiment analysis.

Overall, the type of video scenario exerts a substantial impact on model performance, and enhancing model stability and cross-scenario generalization remains a critical challenge in current research.

## H.2 EFFECT OF NUMBER OF ON-SCREEN SPEAKERS: SINGLE VS. MULTIPLE

In multimodal sentiment analysis, the interaction patterns among speakers constitute a critical factor affecting both task complexity and model performance. As the complexity of interactions increases, the difficulty of the task rises markedly. In this section, we report the accuracies of conventional SOTA approaches and multimodal large language models—specifically Qwen2.5-Omni-7B and MiniCPM-o-2.6—on three-class and five-class sentiment recognition tasks across single-person, multi-person without interaction, and multi-person with interaction scenarios. The evaluations were conducted under random seeds 0–4, as shown in Figure 8.

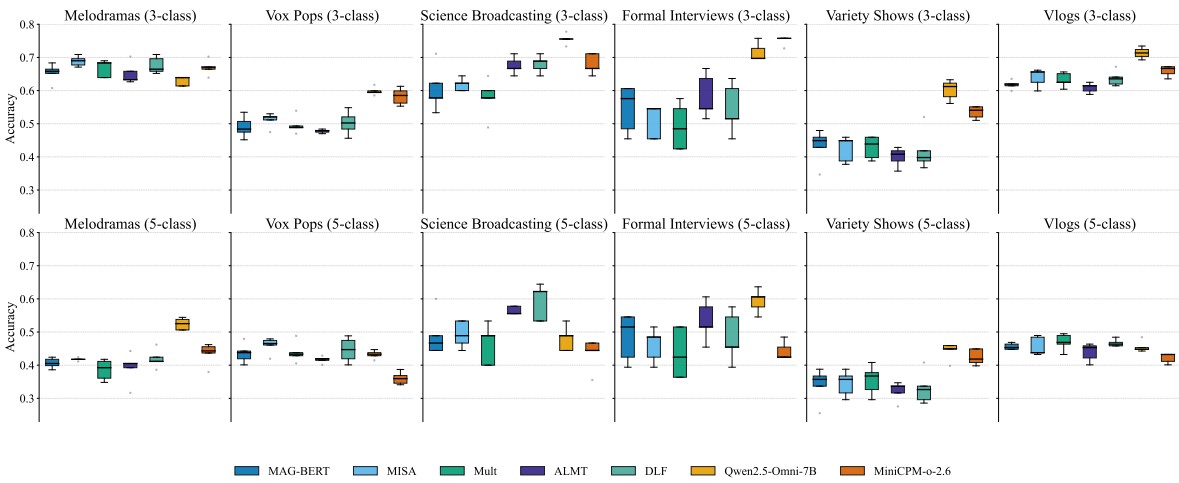

Figure 7: Accuracies of conventional SOTA approaches and multimodal large language models, specifically Qwen2.5-Omni-7B and MiniCPM-o-2.6, for three-class and five-class sentiment recognition across six scenarios—melodramas, vox pops, science broadcasting, formal interviews, variety shows, and vlogs—evaluated under random seeds 0–4.

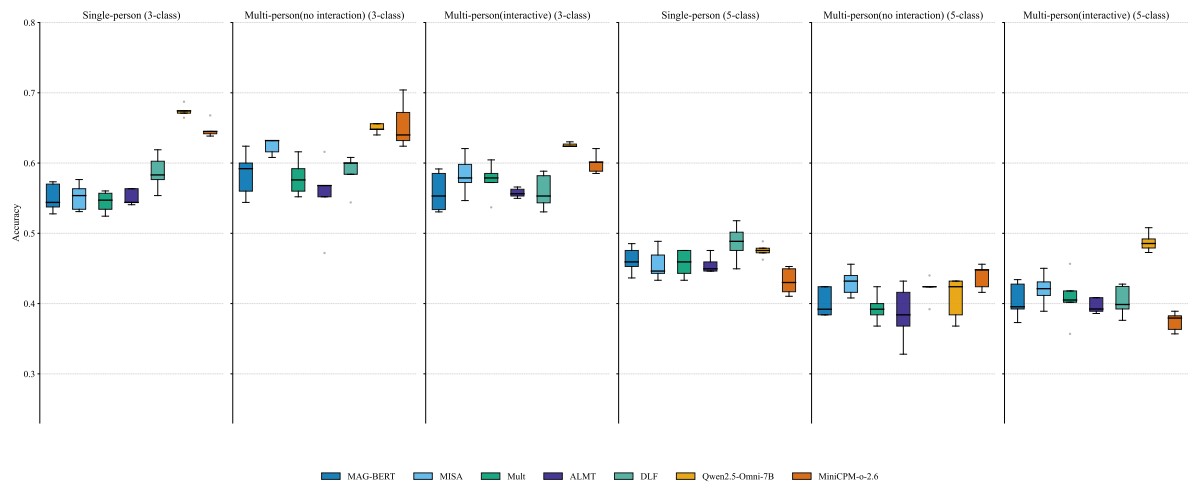

Figure 8: Accuracies of conventional SOTA approaches and multimodal large language models, specifically Qwen2.5-Omni-7B and MiniCPM-o-2.6, are reported for three-class and five-class sentiment recognition across single-person, multi-person without interaction, and multi-person with interaction scenarios, evaluated under random seeds 0–4.

In single-person scenarios, where emotional cues are clear and modal signals are consistent, model recognition is comparatively straightforward, resulting in higher overall accuracy. Small models demonstrate compact accuracy distributions without outliers, reflecting strong stability and low sensitivity to random seeds. By contrast, Qwen2.5-Omni-7B, while generally exhibiting concentrated accuracy distributions, generates a relatively high number of outliers, indicating potential performance variability under certain conditions.In multi-person scenarios without interaction, the model is required to process multiple independent emotional signals simultaneously, thereby increasing recognition difficulty, reducing overall accuracy, and resulting in more dispersed distributions with elongated box ranges. Both the ALMT and DLF methods display outliers, reflecting heightened sensitivity to random seeds, greater training variability, and relatively lower stability.In multi-person scenarios with interaction, the model must capture the dynamic transmission, conflict, and integration of emotions. This requirement imposes higher demands on contextual awareness and relational reasoning, making the task highly complex. In this context, the box range expands significantly. The MULT method exhibits prominent outliers in five-class tasks, accompanied by substantial performance fluctuations, further indicating reduced stability under complex interactive conditions. Nevertheless, Qwen2.5-Omni-7B maintains relatively high accuracy in this scenario, demonstrating its robustness and effectiveness in handling complex contexts.

Furthermore, across different speaker configurations and for both three-class and five-class tasks, MAG-BERT and MISA exhibit no outliers, demonstrating strong robustness, generalization capability, and stability. Overall, three-class tasks consistently achieve higher accuracy than five-class tasks. In three-class settings, large models generally outperform small models, exhibiting higher accuracy distributions. In contrast, in five-class tasks, the advantage of large models diminishes due to finer-grained categories and more ambiguous inter-class boundaries, and in certain cases, they may perform worse than small models.

In summary, with increasing interaction complexity and a greater number of classification categories, model performance exhibits more pronounced fluctuations, accompanied by heightened sensitivity to random seeds. These observations offer valuable insights for model optimization and methodological selection in complex speaker scenarios.

## I  DATA PRIVACY AND CONTENT CONSIDERATIONS

Our dataset has undergone systematic, meticulous screening and includes characters and dialogues from six distinct domains: vox pops, variety shows, melodramas, formal interviews, vlogs, and science-broadcasting programs. Although some material originates from real-world settings, we applied comprehensive anonymization and de-identification procedures to all content involving personal identities to eliminate the risk of privacy breaches and to ensure compliance with applicable privacy laws and regulations. Furthermore, we performed a thorough content review and removed any samples that could be offensive or controversial, thereby guaranteeing social appropriateness. The dataset is intended for the study of character dialogues and interactions, aiming to enable comprehensive, multimodal sentiment analysis through holistic character understanding. The data are used solely for scientific research and do not violate legal rights. All data-processing and research procedures adhere strictly to established ethical guidelines.

## J  LLM USAGE

In our work, both the reasoning process data contribution and methods research use large language model as illustrated in Section 4. In addition to this, we use LLM such as GPT-5 to polish grammar and generate draft icons for figures. All reported experimental results are obtained independently of these editorial aids.

## K  REPRODUCIBILITY

To ensure the reproducibility of the benchmark studies on our dataset, we have provided comprehensive documentation and resources. The data sources, along with the complete data processing, filtering pipeline, and detailed annotation guidelines, are detailed in Appendix A. The experimental setup for the main evaluations, including model implementations, hyperparameters, and evaluation protocols, is elaborated in Section 5. Furthermore, the prompts used for large language model-based experiments are fully listed in Appendix E for reference. The code for data loading, benchmark models, and evaluation, alongside a sample of the processed dataset, is available via an anonymized repository (link provided in the abstract). To account for randomness, we report the standard deviation across multiple random seeds in Appendix G. Upon acceptance of this paper, we commit to releasing the full dataset publicly to further support research in the community.

