# OpenReview forum: "CH-CEMS: A Chinese Multi-Concept Benchmark Dataset Towards Explainable Multi-Modal Sentiment Analysis"
_ICLR.cc/2026/Conference — ICLR 2026 Conference Withdrawn Submission_

### Official Review · Reviewer_Xbm8 · 2025-10-25

**Soundness:** 2
**Presentation:** 2
**Contribution:** 2
**Rating:** 2
**Confidence:** 4

**Summary:**

This paper focuses on explainable multimodal emotion recognition, introducing a dataset (CH-CEMS) and a GRPO-based pipeline to address this task. The key novelty lies in leveraging semantic concepts, where the dataset annotates these concepts and the framework uses their recognition accuracy as rewards during reinforcement learning.

**Strengths:**

1.	A new dataset for explainable multimodal emotion recognition.

2.	Incorporation of semantic concepts during both dataset construction and reward design.

3.	Performance improvements compared to existing MLLMs.

**Weaknesses:**

1.	The paper claims that CH-CEMS is “the first multimodal sentiment dataset for explainable multimodal sentiment analysis” and “the first Chinese dataset for explainable multimodal sentiment analysis”. However, Explainable Multimodal Sentiment Analysis is an active research area with prior datasets (e.g., EMER and MERR) that also provide reasoning cues. Thus, the novelty claim is overstated.

2.	“EMER curates multimodal emotion–reasoning pairs (Liu et al., 2023a) and MERR provides a multimodal emotion description and reasoning benchmark (Zhang et al., 2024). PanoSent introduces a multimodal conversational ABSA benchmark for panoptic sextuple extraction and sentiment ﬂipping with causal rationales (Luo et al., 2024). However, to the best of our knowledge, these resources primarily target emotion recognition or aspect-based sentiment analysis rather than user-centric sentiment reasoning.” => The paper states that EMER and MERR focus on emotion recognition or aspect-based sentiment analysis rather than user-centric sentiment reasoning. However, to the best of my knowledge, these datasets do involve user-centric sentiment reasoning.

3.	The paper classifies semantic concepts into only three categories (speaking style, tone of voice, and facial expression). However, emotion reasoning also depends on other cues (e.g., gestures, events, background, linguistic content). Focusing on such a narrow set may reduce the completeness of emotion reasoning.

4.	As shown in Figures 2 and 4, the authors categorize semantic concepts (e.g., speaking style, tone of voice) into fixed labels. However, these concepts are complex and context-dependent, making a fixed taxonomy potentially inaccurate or incomplete.

5.	Some abbreviations (e.g., SP, SI, EC, adverts in Table 1) are not clearly defined.

6.	There are also citation errors in the paper. Proofreading is recommended to improve clarity and accuracy.

7.	Prior emotion reasoning frameworks (e.g., R1-Omni, Emotion-LLaMA, AffectGPT) already use emotion descriptions containing semantic concepts (e.g., speaking style, tone of voice, facial expression). This paper proposes explicitly extracting these concepts from descriptions. However, it is unclear how this differs from or improves upon traditional description-generation methods. From my perspective, previous description generation methods can cover more semantic concepts but this paper only focuses on limited semantic concepts. I cannot figure out the benefit of this approach. Therefore, a discussion on the advantages of a concept-driven approach is needed.

8.	"Thus We accordingly annotate" → Should be "Thus, we accordingly annotate"(lowercase "we")

**Questions:**

See weakness.

---

> ### Author Response · Authors · 2025-11-21
> **Rebuttal by Authors**
>
> Thank you for the detailed comments. Several concerns seem to come from a misunderstanding of our task formulation; we will revise the paper to make this distinction clearer.
>
> **(1) Task and novelty (sentiment vs emotion).**
> CH‑CEMS follows the multimodal *sentiment* analysis line (CMU‑MOSI/MOSEI, CH‑SIMS), where the target is sentiment polarity and intensity of an opinionated utterance. EMER, MERR, Emotion‑LLaMA, AffectGPT, R1‑Omni and OV‑MER are built for multimodal *emotion* recognition or open‑vocabulary emotion reasoning with discrete/continuous emotion labels and rich emotion descriptions, not sentiment scores. We fully agree that they are important predecessors in explainable affective computing and will make this clearer. Our intended claim is that CH‑CEMS is, to our knowledge, the first **Chinese MSA dataset** that (i) uses the standard MSA sentiment scheme and (ii) adds human‑annotated, categorical concepts plus CoT traces aligned with it. To avoid over‑claiming, we will soften the “first … dataset” wording to “a new Chinese dataset” and explicitly separate “sentiment analysis” from “emotion recognition” throughout the paper.
>
> **(2) On “user‑centric sentiment reasoning”.**
> We did not intend to suggest that EMER or MERR are not user‑centric; they clearly provide user‑oriented explanations in the *emotion* label space. What we meant is that they reason about emotion categories/descriptions, whereas CH‑CEMS reasons about sentiment polarity/intensity with concept tags, and PanoSent targets aspect‑based sentiment toward entities/aspects in a conversation. We will rephrase this paragraph to clarify that our use of “user‑centric sentiment reasoning” specifically refers to sentiment directed at the speaker as a whole person, rather than aspect‑level or entity‑level opinions.
>
> **(3) Choice and coverage of semantic concepts.**
> We agree that many other cues (gestures, events, background, etc.) can influence affective interpretation. In this work, our goal is not to enumerate all possible factors, but to test whether supervising MLLMs with a compact yet semantically rich set of high‑level concepts drawn from different modalities can already benefit explainable multimodal sentiment reasoning. Although we organize our supervision into three fields (speaking style, tone of voice, facial expression), each field contains multiple fine‑grained categories; in many concept‑bottleneck or attribute frameworks, these would be treated as distinct concepts rather than a single coarse label. Our experiments suggest that such structured concept tags provide useful additional supervision beyond standard GRPO. We will clarify this design choice and explicitly state that extending CH‑CEMS with richer annotations, including gesture/background cues and potentially open‑vocabulary concept labels, is important future work. We view such open concepts as complementary to our current discrete tags, which are designed to act as structured, verifiable signals for concept‑guided RL.
>
> **(4) Fixed taxonomy and context‑dependence.**
> These concepts are indeed context‑dependent. Our taxonomies are, however, not ad hoc: speaking style is derived from Biber’s register typology, facial expression from basic‑emotion/FACS categories, and tone from affective/interpersonal tone inventories. Discrete labels are a common compromise in affective computing and are necessary for our reward design, which needs ground‑truth concept tags for the concept reward. In the CoT, models may still mention extra free‑form cues; only the three annotated dimensions are checked by the reward. We will clarify this to address the concern that a fixed taxonomy necessarily ignores contextual subtleties.
>
> **(5) Benefit of concept‑driven supervision vs free‑form descriptions.**
> Frameworks such as Emotion‑LLaMA, AffectGPT and R1‑Omni generate rich emotion descriptions that may implicitly mention style, tone or expression, but these descriptions are not aligned with human concept labels; RL objectives typically supervise only the final prediction or a global quality score. In our framework, the model must output explicit concept tags which are compared to human annotations, and the concept reward penalizes incorrect concepts. This directly constrains the reasoning path and reduces “reward hacking” via unfaithful explanations. Empirically, our concept‑guided GRPO significantly outperforms vanilla GRPO with only format+answer rewards on CH‑CEMS.
>
> We thank the reviewer again for their time and comments, and we will revise the paper accordingly. Some of the concerns, however, seem to arise from factual misunderstandings of our setting (e.g., treating CH‑CEMS as an emotion‑recognition dataset rather than a sentiment‑analysis benchmark dataset and misinterpreting our stated contributions). We hope that, after considering the clarifications and planned revisions above, you may be able to re‑evaluate our work, and we would be happy to engage in further discussion if helpful.

---

### Official Review · Reviewer_dEyJ · 2025-10-27

**Soundness:** 2
**Presentation:** 2
**Contribution:** 2
**Rating:** 2
**Confidence:** 4

**Summary:**

This paper introduces CH-CEMS, a novel Chinese multimodal sentiment analysis dataset designed for explainability. The dataset comprises 3,715 carefully curated video segments that are manually annotated for emotional polarity and intensity, as well as three key semantic concepts: speaking style, tone of voice, and facial expression. These semantic concepts are engineered as explicit reasoning cues to enable process-level supervision of the model's inference, thereby overcoming the scarcity of high-quality data for explainable multimodal sentiment analysis (EMSA). Furthermore, we propose a reinforcement learning framework, GRPO (Guided Reinforcement Learning with Policy Optimization), for Multimodal Large Language Models (MLLMs), which implements concept-level supervision as verifiable rewards.

**Strengths:**

1. CH-CEMS is the first multimodal sentiment dataset for the Chinese language that supports explainability and possesses process-level supervision capability, thereby filling a significant gap in the interpretability of existing Chinese multimodal sentiment analysis datasets.

2. By annotating the three fine-grained semantic concepts—speaking style, tone of voice, and facial expression—as explicit reasoning cues, the dataset enables direct supervision of the model's inference process.

**Weaknesses:**

1. 3715 samples are relatively small for a multimodal dataset used for deep learning benchmark testing. Although high-quality data is crucial, a small sample size may limit the training and generalization ability of large multimodal models (MLLMs). The authors may need to argue in the main text why high-quality annotation of reasoning clues can compensate for the insufficient sample size.

2. The authors need to further argue the sufficiency of these three concepts as 'interpretability'. For example, whether there are other important cultural or social reasoning clues that have not been captured in the Chinese context.

3. The baseline method framework based on SFT (Supervised Fine Tuning) and combined with GRPO (Guided Reinforcement Learning with Policy Optimization) for reinforcement learning proposed by the author is a relatively standard paradigm in current MLLM research. It lacks sufficient novelty and uniqueness at the methodological level.

**Questions:**

1. The paper provides kappa scores for Tone, Expression, Speaking Style, and sentiment, but they are all between 0.45-0.55, indicating a moderate level of annotation consistency. Therefore, there is controversy over the paper's claim that this dataset is of high quality. A high score of Inter-Annotator Agreement is the key to successful process level supervision.

2. Please provide more detailed statistical information on data distribution in the paper. For example, the category distribution and co-occurrence of three semantic concepts (style, intonation, expression) (such as whether a certain "speaking style" is always accompanied by a certain "facial expression").

3. In Section 5, was the performance difference of the model in emotion prediction and explanatory generation discussed when using and not using these three explicit inference clues for process supervision? If not, this should be presented as a core experimental result to highlight the unique value of the CH-CEMS dataset.

4. Why do you use feature-based methods only to conduct regression task? What about LLMs?

---

### Official Review · Reviewer_QAN3 · 2025-10-31

**Soundness:** 3
**Presentation:** 3
**Contribution:** 3
**Rating:** 6
**Confidence:** 5

**Summary:**

The paper proposes multi-concept datasets for explainable multimodal sentiment analysis task and conduct concept-based GRPO with effective fine-grained reasoning process. Experiments on MOSI/MOSEI are conducted to show effectiveness of the proposed model.

**Strengths:**

1.	Comprehensive dataset.
2.	Experiments are sufficient.

**Weaknesses:**

1.	More recent baseline models with small model such as RoBERTa, DeBERTa should be conducted to show fair comparison.
2.	Multi-person cases should be provided.
3.	The SFT version of the proposed model should be included for comparison. Cold start performance should be reported to better show multi-stage training efficiency.
4.	The most number of neutral class may reduce the effectiveness of the presented dataset.

**Questions:**

1.	How to ensure the correctness of reasoning process? Any human value alignment stage in annotation?
2.	Why does GRPO not help for Qwen2.5-Omni compared with SFT version?
3.	Since the description of concept could be different with similar semantics, does reward computation considering such situations?
4.	Why does the neutral class emotion with 0 regressive value have the most numbers, much more than other classes which show intensity of emotion?

**Details Of Ethics Concerns:**

Are the human privacy in all videos guaranteed?

---

### Official Review · Reviewer_yWf7 · 2025-10-31

**Soundness:** 3
**Presentation:** 2
**Contribution:** 2
**Rating:** 4
**Confidence:** 3

**Summary:**

The paper presents CH-CEMS, a new Chinese multimodal dataset (text-audio-video) for explainable sentiment reasoning. Each clip (3,715 total) is annotated with sentiment polarity, intensity, and three intermediate concepts — speaking style, tone of voice, and facial expression — plus corresponding chain-of-thought traces.
The authors also propose a concept-guided GRPO (Group Relative Policy Optimization) framework that rewards models for correct structure, sentiment prediction, and concept alignment. Experiments are conducted on MLLMs and feature-based baselines, reporting up to 73.6% accuracy on 3-class sentiment classification, claimed to surpass several closed-source large models.

**Strengths:**

Process-level supervision: Incorporating three interpretable concept dimensions (style, tone, expression) provides more transparent reasoning supervision than typical black-box multimodal sentiment datasets.

Methodical RL extension: The concept-guided GRPO is a reasonable and reproducible adaptation of RLHF/GRPO to emotion reasoning, with clear reward decomposition.

Transparency: The paper provides dataset splits, inter-annotator statistics, prompts, and ethical disclosure, which improve reproducibility.

Timely topic: Explainable multimodal emotion reasoning is increasingly important in the era of MLLMs and aligns well with ICLR’s focus on interpretable and grounded learning.

**Weaknesses:**

1. Motivation and “Why Chinese?”

The motivation for focusing on Chinese data remains weak.
While the authors argue that Chinese multimodal sentiment data are under-represented, there is no empirical demonstration that Mandarin introduces unique challenges (e.g., tonal prosody, discourse politeness, idiomatic affect) that justify a separate dataset.
Without comparative cross-lingual experiments (e.g., transfer CH-CEMS to other commonly known datasets), the argument reads as geographical novelty rather than scientific necessity.

2. Dataset Soundness

The dataset’s size (3,715 clips) is modest for training MLLMs and may limit generalization. The total length is only 7 minutes, which is confusing. Inter-annotator κ ≈ 0.46 – 0.54 indicates moderate agreement; noise at this level restricts the theoretical ceiling of achievable accuracy.

3. . Evaluation Scope

Evaluation is limited to the authors’ dataset; there are no cross-dataset generalization tests (e.g., MER, EmoSet). Faithfulness of the generated reasoning is measured only via concept match, not by human or counterfactual verification.

4. Missing Key Baselines and Model Comparisons

The literature review is significantly incomplete, omitting all major recent models in multimodal emotion reasoning: EmoVIT (CVPR 2024) – a unified visual-language transformer with affective grounding, establishing a strong multimodal benchmark. AffectGPT (ICML 2025) – a generative multimodal LLM fine-tuned for emotion reasoning and affective dialogue understanding. EmoSet (ICCV 2023) – employs attribute-based reasoning on a large-scale fine-grained emotion dataset, conceptually similar to this work. OV-MER (ICML 2025) – supports open-vocabulary multimodal emotion reasoning with > 200 continuous labels and powers the MER 2025 Challenge benchmark. Without discussing or comparing against these, the paper’s claimed novelty in concept-guided multimodal emotion reasoning is undermined. For ICLR, inclusion of these baselines or at least a thorough discussion is mandatory.

Others:

✓ symbols missing: In Table 1, the check marks indicating modality and concept availability are incomplete.
The number format in the table is not consistent.
Typos can be observed, e.g., in 214.

**Questions:**

In general, this is a good paper, but the authors need to address two main concerns before it can reach the bar of ICLR. 1. It is necessary to prove the generalizability of the proposed method on the language aspect (not limited to Chinese). 2. It's vital and beneficial to verify the trained model on other existing datasets, to show the gain of introducing reasoning and multi-attribute in the emotion understanding task.

I will adjust the rating accordingly.

---

> ### Author Response · Authors · 2025-11-21
> **Rebuttal by Authors**
>
> Thank you for the constructive review. We address each concern below.
>
> 1. Motivation and “Why Chinese?”
>    Our focus on Chinese is due to a resource gap and practical considerations, while the method itself is language‑agnostic.
>    (1) Resource gap. Most explainable affective datasets are for emotion recognition, not *sentiment* (polarity + intensity). CH‑CEMS is, to our knowledge, the first Chinese multimodal dataset that jointly provides (i) sentiment polarity + intensity, (ii) three interpretable concepts (style, tone, expression), and (iii) chain‑of‑thought traces for sentiment reasoning.
>    (2) Linguistic relevance, but not limitation. The three concepts are defined from general linguistic/psychological theory and can be instantiated in any language. Chinese is a natural first testbed because tonal prosody and discourse conventions distribute sentiment cues across prosody, facial expression and speaking style beyond lexical polarity.
>    (3) Language‑agnostic method. Concept‑guided GRPO only uses rewards for (i) template correctness, (ii) sentiment label, and (iii) concept labels; no Chinese‑specific heuristics are used. The backbone (Qwen2.5‑Omni‑7B) is multilingual, and our prompts/tag schema can be translated. In addition, we are setting up small cross‑lingual experiments by applying the trained model to translated non‑Chinese samples and will report results if ready.
>
> 2. Dataset soundness (size, duration, agreement)
>    (1) Size: We agree 3,715 clips is modest for pre‑training, but CH‑CEMS is used only for post‑training. Each clip contributes one utterance (~35 tokens), three concept labels, and a calibrated reasoning trace, giving much denser supervision than standard single‑label MSA datasets. This proved sufficient for GRPO to improve a strong multilingual backbone over both open‑ and closed‑source models on CH‑CEMS.
>    (2) Duration. “07:54” in Table 1 denotes **7 hours 54 minutes**, not 7 minutes 54 seconds. We will change the notation to “7h54m” and clarify in the caption.
>    (3) Inter‑annotator agreement. Fleiss’ κ≈0.46–0.54 indicates moderate agreement, common for fine‑grained subjective affect labels but still a limitation. To reduce noise we used five annotators, required 3‑of‑5 agreement with re‑annotation when necessary, and averaged scalar sentiment scores before mapping to 3/5‑class labels. We will clarify this pipeline and note that we are extending the dataset and applying stricter adjudication to further improve reliability.
>
> 3. Evaluation scope and reasoning faithfulness
>    (1) Cross‑dataset / cross‑task evaluation. We did not include MER/EmoSet/OV‑MER/AffectGPT experiments because their label spaces and goals differ from CH‑CEMS: they target discrete or open‑vocabulary emotions, often with free‑form descriptions, whereas we target sentiment polarity + intensity plus three specific concepts. These datasets do not provide categorical labels for style/tone/expression, so using our concept reward would require extra human concept annotation or a new reward based on textual descriptions/attributes. We will make this limitation explicit and, if time permits, add small‑scale transfer experiments with a relaxed reward that ignores concept labels.
>    (2) Faithfulness of reasoning. In the main text we use concept‑match as an automatic proxy: the concept reward enforces consistency between predicted concept tags and human labels, discouraging obviously inconsistent explanations. We agree this does not fully capture natural‑language quality or global faithfulness. During the rebuttal period we plan to run an additional evaluation using GPT‑4 to compare vanilla GRPO and concept‑guided GRPO on coherence and consistency with multimodal evidence, and will incorporate the results if completed.
>
> 4. Missing baselines and model comparisons
>    Thank you for pointing out EmoVIT, AffectGPT, EmoSet, and OV‑MER. Our experiments were organized around MSA baselines (MuLT, MISA, ALMT, etc.) plus strong MLLMs (Qwen2.5‑VL/Omni, GPT‑4o, GPT‑5, Gemini‑2.5 Pro). We will add a paragraph in Related Work to situate these methods: EmoSet/EmoVIT are visual emotion benchmarks/models; AffectGPT, EMER, OV‑MER and similar systems address discrete or open‑vocabulary emotion recognition and description. In contrast, CH‑CEMS is a multimodal *sentiment* benchmark with polarity + intensity and three explicit concepts, plus chain‑of‑thought traces for process‑level supervision. Because of this different task and label space, these methods are not directly comparable as numerical baselines on CH‑CEMS, but we will clarify their relationship and emphasize that our concept‑guided GRPO could in principle be combined with such architectures if compatible concept‑level supervision were available.

---

### Note · Authors · 2025-12-08

I have read and agree with the venue's withdrawal policy on behalf of myself and my co-authors.